# Rapid adaptation to a novel pathogen through disease tolerance in a wild songbird

**Amberleigh E. Henschen**[1,2]*, **Michal Vinkler**[3], **Marissa M. Langager**[4], **Allison A. Rowley**[4], **Rami A. Dalloul**[5], **Dana M. Hawley**[4], **James S. Adelman**[1,2]*

**1** Department of Biological Sciences, University of Memphis; Memphis, Tennessee, United States of America, **2** Department of Natural Resource Ecology and Management, Iowa State University; Ames, Iowa, United States of America, **3** Department of Zoology, Charles University; Prague, Czech Republic, **4** Department of Biological Sciences, Virginia Tech; Blacksburg, Virginia, United States of America, **5** Department of Poultry Science, University of Georgia; Athens, Georgia, United States of America

* aehenschen@gmail.com (AEH); jim.adelman@memphis.edu (JSA)

**Data Availability Statement:** The data that support the findings of this study are publicly available from

## Abstract

Animal hosts can adapt to emerging infectious disease through both disease resistance, which decreases pathogen numbers, and disease tolerance, which limits damage during infection without limiting pathogen replication. Both resistance and tolerance mechanisms can drive pathogen transmission dynamics. However, it is not well understood how quickly host tolerance evolves in response to novel pathogens or what physiological mechanisms underlie this defense. Using natural populations of house finches (*Haemorhous mexicanus*) across the temporal invasion gradient of a recently emerged bacterial pathogen (*Mycoplasma gallisepticum*), we find rapid evolution of tolerance (<25 years). In particular, populations with a longer history of MG endemism have less pathology but similar pathogen loads compared with populations with a shorter history of MG endemism. Further, gene expression data reveal that more-targeted immune responses early in infection are associated with tolerance. These results suggest an important role for tolerance in host adaptation to emerging infectious diseases, a phenomenon with broad implications for pathogen spread and evolution.

## Author summary

When organisms encounter a novel pathogen, they can adapt to it in two ways: by killing the pathogen (known as resistance) or reducing the damage incurred while not directly killing the pathogen (known as disease tolerance). Although much work has focused on resistance, we understand less about how animals achieve disease tolerance or how quickly disease tolerance can evolve. Here we show that a wild bird species (the house finch) has evolved disease tolerance to a novel bacterial pathogen quickly (within ˜20–25 years, at most ˜15 generations). In house finches, this pathogen causes severe swelling around the eye and limits their ability to avoid predators. House finch populations that have evolved with this pathogen for longer are more tolerant to it; they show milder eye swelling even though they do not clear the pathogen any more efficiently than their less-tolerant

Dryad Digital Repository (https://doi.org/10.5061/dryad.v9s4mw71r) and NCBI (BioProject ID: PRJNA973136; https://www.ncbi.nlm.nih.gov/bioproject/PRJNA973136).

**Funding:** Funding for this project was provided by National Science Foundation (nsf.gov) grants 1950307 (JSA) and 1754872 (RAD, DMH). The funders had no role in study design, data collection and analysis, decision to publish, or preparation of the manuscript.

**Competing interests:** The authors have declared that no competing interests exist.

counterparts. Moreover, tolerant finches have fewer immune genes that turn on in response to the infection, suggesting that a more-targeted immune response may facilitate tolerance. As disease tolerance can potentially help humans and animals adapt to new pathogens and also may change the way pathogens spread through animal populations, it is critical we understand how, and how quickly, tolerance evolves.

## Introduction

Animal hosts face intense selective pressure to limit morbidity and mortality from emerging pathogens [1–3]. To combat novel pathogens, host populations can rapidly evolve the ability to resist infection by effectively killing and controlling replication of these pathogens. For example, recent evidence suggests that resistance evolved rapidly to emerging fungal pathogens of amphibians [4] and bats [5], as well as myxoma virus in rabbits [6]. However, it is less clear whether disease tolerance, which limits the damage incurred during infection without necessarily reducing pathogen replication [7–9], can evolve on similar timescales among natural populations confronted with novel pathogens [10]. This is important as, depending on the underlying mechanisms (e.g., whether pathology during infection facilitates or hinders pathogen transmission), tolerance could either increase or decrease host competence to transmit pathogens relative to resistance [11,12]. Recent work in wild populations of amphibians [13], birds [14], and fish [15] suggests that tolerance may be a widespread strategy facilitating host persistence when new pathogens emerge. While this work has focused on host survival in response to pathogen or parasite infection, we currently lack empirical evidence of how tolerance changes in animal populations with time since a pathogen's emergence. Further, few studies have probed the potential mechanisms that might underlie this evolution in natural populations. This work is key for predicting the impacts of host tolerance on pathogen transmission and evolution.

House finches (*Haemorhous mexicanus*) and their recently emerged bacterial pathogen (*Mycoplasma gallisepticum*, MG) are a promising host-pathogen system in which to explore the evolution and potential mechanisms of disease tolerance. Originally a pathogen of poultry, MG emerged in house finches in the early 1990s in the mid-Atlantic, USA [16]. In house finches, MG causes severe conjunctivitis [16,17] and decreases survival in the wild [18]. After spreading westward through the eastern and northwestern USA, MG recently moved into some southwestern USA populations [19,20]. Thus, MG endemism in house finch populations ranges from ˜25 years (e.g., in Virginia) to <10 years (e.g., in southern Arizona), with MG not yet detected in some parts of the southwest (e.g., northern Arizona) or the Hawaiian Islands (Fig 1) [16,20–27]. This allows for a space-for-time approach to determine if MG endemism is related to tolerance across populations. Evidence from the wild suggests that MG exerts strong selection pressure on house finch populations, with losses from infection-mediated mortality reducing populations to about 40% of pre-MG numbers within three years of pathogen arrival [28]. Mortality risk in house finches infected with MG appears related to expressing conjunctivitis during infection [18], which reduces the ability of individuals to avoid predators [29]. Thus, selection should favor host responses that limit conjunctival swelling and the associated mortality during MG infection. Indeed, there is considerable evidence that house finches have evolved to adapt to MG [10,30–32]. Although this previous work suggests population differences in both resistance and tolerance to MG, these studies included only one or two populations and thus cannot definitively address whether such patterns reflect pathogen-mediated selection, rather than drift or selection based on differences in other myriad biotic and abiotic factors [33].

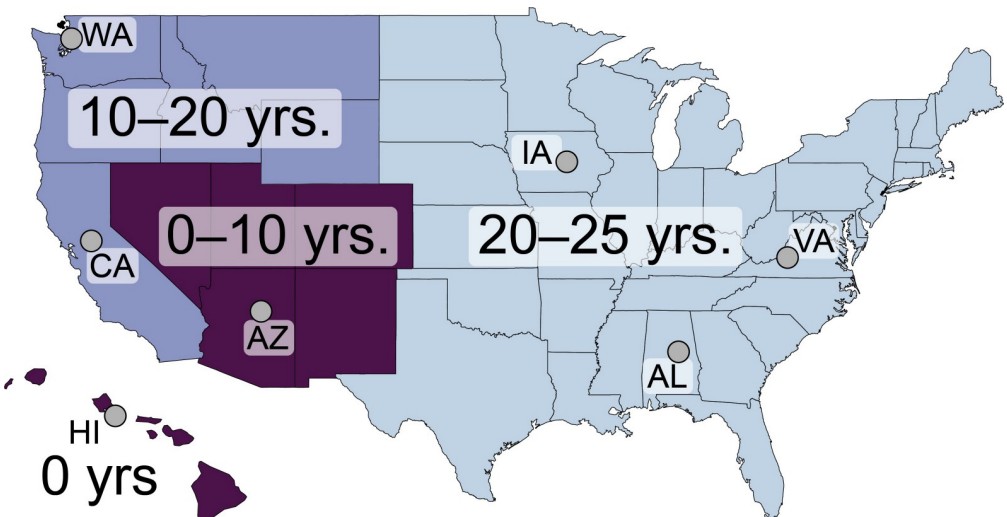

**Fig 1. Using a captive experimental approach, we assayed house finches from seven populations (grey circles) spanning the temporal invasion gradient of the bacterial pathogen, *Mycoplasma gallisepticum* (MG).** From oldest to most recent, these are: Virginia (VA), Alabama (AL), Iowa (IA), Washington (WA), California (CA), Arizona (AZ), and Hawaii (HI). This space-for-time approach leverages the well-documented spread of MG eastward from VA beginning in the early 1990s. After reaching the northwestern US around 2004 (e.g., WA), MG spread along the western coast (CA) and into some populations in the desert southwest within the last 0–10 years (e.g., AZ). MG has not been detected in finches in HI [16,19–27]. Use of line drawing maps created using the maps package in R is allowed under a general public license: https://cran.r-project.org/web/licenses/GPL-2.

In addition to limited empirical data on the evolution of tolerance in house finches, the underlying mechanisms of tolerance remain largely unknown. Given the nature of MG pathology (i.e., conjunctivitis) in house finches, inflammatory pathways are likely an important target of selection for tolerance to MG. Indeed, previous gene expression assays showed that house finches from a more-tolerant population had a lower ratio of pro- to anti-inflammatory cytokine expression (IL-1β:IL-10) early in MG infection, when compared with individuals from a less-tolerant population [31]. There is also evidence that dampening inflammatory immune responses [15,34] and a balanced expression of pro- and anti-inflammatory cytokines [35] may increase tolerance to infection in other species. However, mechanisms of tolerance can also involve myriad other physiological or behavioral changes [14,36–38]. Transcriptome profiles of infected individuals from populations that differ in tolerance can help address this challenge by assessing gene expression simultaneously across many diverse pathways.

In this study, we used a space-for-time approach to test for host evolution after the emergence of a novel pathogen (MG) in a wild bird species (house finches). We experimentally assayed tolerance, using differences in patterns of pathology versus pathogen load, in MG-naïve juvenile birds from seven house finch populations that span the temporal invasion gradient of MG (Fig 1): three populations where MG has been endemic for 20–25 years (Virginia, Alabama, Iowa), two populations where MG has been present for 10–20 years (Washington, California), and two populations where MG has been present for less than 10 years or has never been detected (Arizona, Hawaii; Fig 1) [16,20–27]. This allowed us to account for the number of years over which pathogen-mediated selection has potentially acted (i.e., pathogen pressure), while accounting for other biotic and abiotic pressures that differ spatially [39]. After birds were brought into a common, captive environment, we collected pre-inoculation samples from all individuals, and then conducted three experiments to assess variation in host tolerance and its underlying mechanisms across populations.

In the first experiment, we inoculated birds with either a low or high dose of an evolutionarily basal, lower-virulence MG isolate (Table 1). To ensure that our results were not unique to a single pathogen isolate [40], in the second experiment we used a more evolutionarily derived, higher-virulence MG isolate, at a single, low dose (Table 1). In the third experiment, we used transcriptomics to quantify gene expression three days post-inoculation in a secondary lymphoid organ associated with the eye (the Harderian gland) [41]. We compared gene expression in non-infected and MG-infected finches from two more-tolerant (IA, VA) and two less-tolerant (AZ, HI) populations (Table 2, see Results). This allowed us to better understand the roles of genes involved in inflammatory pathways and other physiological processes in tolerance to MG. To best isolate potential mechanisms of tolerance per se, here we used birds inoculated with a high dose of a lower-virulence isolate, which had revealed strong population differences in tolerance, but not resistance (see Results). We predicted that populations in which MG has been endemic longest would have 1) the highest tolerance to both the basal, lower-virulence isolate and the more derived, higher-virulence isolate; 2) a larger percentage of asymptomatic infections (i.e., the most extreme form of tolerance in which individuals have measurable pathogen load but no conjunctivitis during infection); and 3) fewer upregulated genes that code for pro-inflammatory proteins.

## Methods

### Ethics statement

All animal procedures were approved by Institutional Animal Care and Use Committees (IACUC) at Iowa State University (ISU), The University of Memphis (UM), and Virginia Tech (VT), as well as the ISU Institutional Biosafety Committee. We were permitted by appropriate state and federal agencies to work with wild songbirds (Table A in S1 Appendix).

### Capture, transportation, and housing

Using both mist nets and feeder traps, we captured juvenile house finches (aged by plumage characteristics) [42] between June and September 2018 in Blacksburg, Virginia; Ames, Iowa; Tempe, Arizona; and Waianae, Oahu, Hawaii. Between July and August 2019, we captured house finches in Bothell, Washington; Davis, California; and Auburn, Alabama (GPS coordinates of capture sites in Table B in S1 Appendix). We immediately released any finches showing clinical signs of MG infection at the time of capture.

After capture, we fitted each individual with a uniquely numbered aluminum leg band and measured their mass using an electronic balance. We then dusted birds with 5% Sevin powder to eliminate ectoparasites. We transferred all birds to ISU (Ames, IA, 2018), UM (Memphis, TN, 2019), or VT (Blacksburg, VA, 2018–2019) by car or a combination of car and airplane in

**Table 1. Experimental design for the longitudinal infection studies (Experiment 1 and 2), where birds from each population were randomly assigned to one of three inoculation treatments.** We collected pre-inoculation samples from all birds, which served as controls. Sample sizes are not even across dose / isolate treatments due to the ethical and logistical limitations of working with wild vertebrates from seven geographically separated populations.

| | Years MG Endemic | | | | | | |
|---|---|---|---|---|---|---|---|
| | 20–25 | | | 10–20 | | 0–10 | |
| | Population | | | | | | |
| Treatment | AL | IA | VA | CA | WA | AZ | HI |
| Lower-virulence isolate, Low dose | 10 | 10 | 10 | 10 | 10 | 10 | 11 |
| Lower-virulence isolate, High dose | 10 | 10 | 10 | 9 | 10 | 10 | 11 |
| Higher-virulence isolate, Low dose | 10 | 11 | 10 | 10 | 0 | 10 | 10 |

**Table 2. Summary of treatments for the gene expression study (Experiment 3).** In this study, we used a subset of four populations representative of the extremes of MG endemism and tolerance (two "Less-tolerant", two "More-tolerant"). Harderian glands were collected from all individuals three days post-inoculation with *M. gallisepticum*. *The sample from one bird from HI showed a near zero read-count after sequencing and was excluded from subsequent analyses.

| Populations | Treatment | Replicates |
|---|---|---|
| Less-tolerant (AZ, HI) | Negative control | 10* |
| | Lower-virulence isolate–High dose | 10 |
| More-tolerant (VA, IA) | Negative control | 10 |
| | Lower-virulence isolate–High dose | 10 |

IATA-approved pet carriers modified for birds. All birds went through an acclimation and quarantine period after arrival (minimum of 40 days), including treatment with prophylactic medications to prevent disease from natural pathogens (*Quarantine and prophylactic medications* section of S1 Appendix). Individuals showing any sign of prior exposure to MG, including pathology or anti-MG antibodies (see below) were not included in the experiments. During our experiments, we housed birds singly in medium flight cages (76 cm x 46 cm x 46 cm), and provided *ad libitum* water and food, consisting of a 20:80 mix of black oil sunflower seed:pellets (Roudybush Maintenance Nibles; Roudybush, Inc., Woodland, CA). We held light and dark cycles (12h:12h) and temperatures (~22°C) constant.

## Experimental infections: longitudinal and RNA-seq studies

For Experiments 1 and 2 (i.e., the longitudinal studies), we sampled all animals 9 or 10 days prior to experimental infection, so each bird served as its own control (see below). We inoculated all birds with 35 μL of Frey's medium containing MG into both conjunctivae. For Experiment 1, birds were randomly assigned to one of two MG treatments (Table 1): 1) low dose ($7.5 \times 10^2$ color changing units, CCU/mL) of a lower-virulence MG isolate (VA94 [7994–1 (6P) 9/17/2018]) [43] or 2) high dose ($7.5 \times 10^6$ CCU/mL) of the same lower-virulence isolate. For Experiment 2, birds were given a low dose ($7.5 \times 10^2$ CCU/mL) of a higher-virulence isolate (VA13 [2013.089–15 (2P) 9/13/2013]) [40]. Due to sample size constraints, we did not inoculate any individuals from Washington with the higher-virulence isolate (Experiment 2), but all other populations were represented in all treatments. We quantified mass, pathogen load, and conjunctivitis (see below for further details) on days 9 or 10 before inoculation and on days post inoculation (DPI) 3, 7, 14, 21, 28, and 34. To quantify anti-MG IgY levels, we collected blood samples (~100 μL) before inoculation, as well as on DPI 14 and 28.

For Experiment 3, the RNA-seq study (which required tissue harvesting post-euthanasia on DPI 3), we inoculated birds from only Virginia, Iowa, Arizona, and Hawaii with either a control treatment (sterile Frey's medium) or the high dose of the lower-virulence isolate (n = 5/population; Table 2). We used the high dose only to maximize our ability to detect transcriptomic responses associated with tolerance versus resistance (see Fig 2). We quantified mass, pathogen load, and conjunctivitis before inoculation and on DPI 3 just prior to euthanasia.

The longitudinal experiments were split across three institutions: ISU (2018), UM (2019), and VT (2018–2019). Birds from IA, VA, AZ, and HI were split evenly across ISU and VT in 2018 and birds from AL and CA were split evenly across UM and VT in 2019. Birds from WA were exclusively assayed at UM in 2019. We held all treatments and sampling procedures constant across institutions and years. Institution and year had no detectable effects on either pathogen load or pathology in the longitudinal experiments (Tables C and D in S1 Appendix).

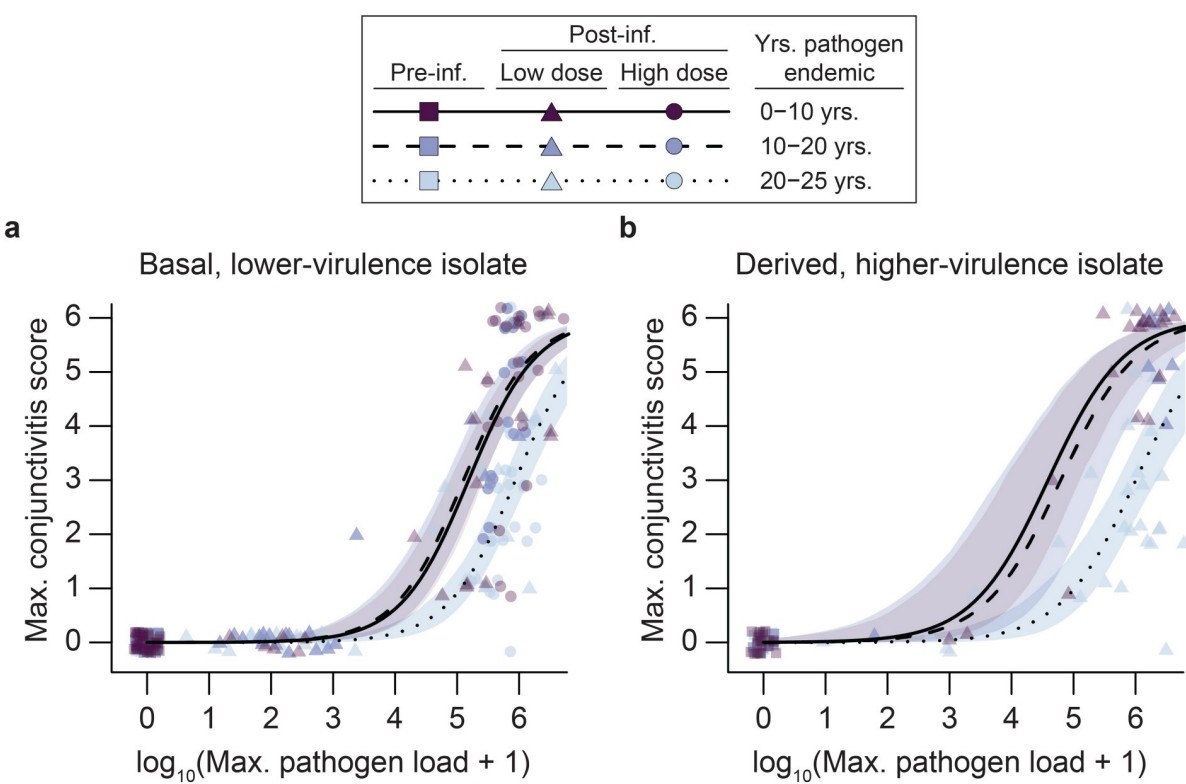

**Fig 2. House finch populations (n = 2–3 populations per endemism category) in which the bacterial pathogen, *Mycoplasma gallisepticum* (MG) has been endemic longer (20–25 years, lighter symbols and shading, dotted line) were more tolerant of experimental infection than house finch populations with little or no history of MG endemism with both (a) an evolutionarily basal, lower-virulence MG isolate (n = 141 birds from 7 populations) and (b) an evolutionarily derived, higher-virulence isolate (n = 61 birds from 6 populations).** We quantified tolerance as the ability to maintain health (y-axis, lower maximum conjunctivitis scores) despite increasing pathogen loads (x-axis; $\log_{10}$(maximum pathogen load + 1)). The most pronounced difference we detected in resistance, measured as lower $\log_{10}$(maximum pathogen load + 1), was within the low-dose, low-virulence treatment, between populations with 0–10 years of pathogen endemism (dark triangles) and those with 10–20 years of pathogen endemism (medium-shaded triangles). For the low-virulence isolate (a), we used data collected from the same individuals both pre-infection and at post-infection peaks in pathogen load and conjunctivitis scores following either a low- or high-dose inoculation ($7.5 \times 10^2$ and $7.5 \times 10^6$ CCU/mL, respectively). For high-virulence infections (b), we performed analyses similarly, but used only the $7.5 \times 10^2$ CCU/mL inoculation dose (see main text). Lines show predictions from generalized linear mixed effects models with shading representing bootstrapped 95% confidence intervals. Data have been jittered in the graph to clearly visualize all points.

We performed all animal procedures for the RNA-seq experiment (Experiment 3) at ISU in 2018.

## Eye scores (conjunctivitis)

Two observers, blind to population origin and treatment (JSA at ISU and UM, DMH at VT), scored conjunctivitis in each eye on a scale of 0–3 at 0.5 intervals. A score of zero denotes an eye that is normal and healthy, one indicates minor eye swelling, two indicates moderate swelling or mild conjunctival eversion, and three denotes severe swelling and conjunctival eversion and exudate [44,45]. For each sampling date, we combined eye scores from the left and right eye, giving each individual a score of 0–6. We also used ImageJ (https://imagej.nih.gov/ij/) [46] to measure the total area of pathology surrounding each eye using pictures taken of each finch in a standardized position against a size marker (for more details see S1 Appendix). These measures of pathology were highly correlated with eye scores given for both the left eye ($r_{423} = 0.79$, $p < 0.001$) and right eye ($r_{429} = 0.83$, $p < 0.001$) (Fig A in S1 Appendix).

## Pathogen load

We quantified pathogen load via qPCR targeting the *mgc*2 gene [47] (PCR details in *Pathogen load quantification* section of S1 Appendix). We swabbed each conjunctiva with a sterile cotton swab dipped in tryptose phosphate broth (TPB) and froze all samples prior to DNA extraction (further details in S1 Appendix).

## Tissue collection, RNA extraction and sequencing

During 2018, we collected Harderian glands from 40 birds on DPI 3 (10 each from AZ, HI, IA, and VA) for Experiment 3. In birds, the Harderian gland, a structure found primarily in animals with nictitating membranes, is populated with lymphocytes and functions as part of the eye-associated lymphoid tissue [41]. We collected tissues within 20 minutes of euthanasia by rapid decapitation and immediately transferred Harderian glands into RNA*later* (Cat. #: AM7021, Thermo Fisher Scientific, Waltham, MA, USA). We stored these samples for 24 hrs at 4˚C and then at -80˚C until RNA extraction. For extraction, we homogenized the tissues using a TissueLyser operated for four minutes at 25 Hz. We extracted RNA with a Qiagen RNAeasy kit (Cat #: 74004; Qiagen, Hilden, Germany), using manufacturer's instructions, including the on-column Dnase procedure.

Before sequencing, we determined the quality of our RNA using an Agilent TapeStation. Our samples had an average RNA integrity number (RIN) of 9.5 (range = 7.7–10), indicating that the RNA was of high quality. Libraries of cDNA were created from mRNA using a TruSeq RNA Library preparation kit (Illumina, San Diego, CA, USA). All Harderian gland samples were individually barcoded and subsequently pooled and run on four lanes of an Illumina NovaSeq 6000 (S4 2X150-bp) platform. After adapters and low quality sections were trimmed using Trimmomatic [48], samples had an average read length of 126–144 bases (average = 136.4; median = 136). Reads per sample ranged from 84 million–151.5 million (average = 107.9; median = 104.4). A single Harderian gland sample from a non-infected control (Hawaiian population) had a near zero read count and was thus excluded from all analyses. Sequencing data used in this study are located at http://www.ncbi.nlm.nih.gov/bioproject/973136 [49].

## De novo transcriptome assembly

We assembled a *de novo* house finch transcriptome using Trinity [50–51]. We used all reads from Harderian gland samples, as well as spleen samples collected from the same individuals (collected, extracted, and sequenced as described for Harderian glands above) to develop the transcriptome. Before assembly, we normalized reads using Trinity's *in silico* read normalization function. This transcriptome had a total of 1,047,427 scaffolds (scaffold size range: 169–69,878) and had an N50 value of 4494. We used BUSCO V4.0.5 Aves database to quantify the completeness of our transcriptome [52]. Out of 8,338 orthologs, 7,067 were present in our transcriptome, while 481 were fragmented, and 790 were missing. Thus, our transcriptome is of similar quality to other recently published avian transcriptomes [53,54]. After assembly, we mapped transcripts (n = 532,002) to our transcriptome using Bowtie2 [55] and quantified transcript abundance using RSEM [56]. We filtered out transcripts if they had both a low number of reads in all samples (<5) and were expressed in fewer than two individuals in all four population and treatment groups (i.e., non-infected birds from less-tolerant populations, infected birds from less-tolerant populations, non-infected birds from more-tolerant populations, infected birds from more-tolerant populations). This filtering reduced the number of unique transcripts (genes and their isoforms) to 15,400.

## Differential gene expression analysis

After filtering, we used edgeR [57] to estimate differential expression of transcripts between infected and non-infected individuals from the same population. We compared non-infected birds from less-tolerant populations with infected birds from less-tolerant populations and non-infected birds from more-tolerant populations with infected birds from more-tolerant populations. This allowed us to control for baseline population differences in gene expression between less- and more-tolerant populations. We considered transcripts to be differentially expressed between groups if they had a false discovery rate (FDR) below 0.05. We followed the Trinotate protocol for functional annotation of transcripts [58]. We then used g:Profiler [59,60] to perform enrichment analyses on differentially expressed gene sets using the g:GOSt tool (options set to 'only annotated genes', 'g:SCS threshold', 'user threshold = 0.05'). We used the 'ordered query' option and ordered genes in each set based on log Fold Change (compared to non-infected individuals). Before enrichment analyses, we split genes into four gene sets: upregulated genes in less-tolerant populations, downregulated genes in less-tolerant populations, upregulated genes in more-tolerant populations, and down-regulated genes in more-tolerant populations. We used these analyses to look for enriched terms for both less-tolerant and more-tolerant populations across a number of databases: Gene Ontology (GO) Molecular Function (MF), Biological Process (BP), and Cellular Component (CC) [61,62], Kyoto Encyclopedia of Genes and Genomes (KEGG) [63–65], Reactome (REAC) [66], WikiPathways (WP) [67], and TRANSFAC (TF) [68]. A list of significantly enriched Gene Ontology Biological Process terms are in Table E in S1 Appendix and a full list of all significantly enriched terms from all databases can be found in S1 Data.

## Statistical analysis of tolerance and resistance

To test how tolerance and resistance varied with years since MG endemism, we constructed linear mixed effects models using the lme4, lmerTest, car, and merTools packages in R (R development core team) [69–73]. All $p$-values reflect two-tailed tests. Effects of variables in final mixed effects models were assessed using either type III sums of squares and Satterthwaite's method (linear mixed effects models) [70] or type II sums of squares and Wald $\chi^2$ tests (generalized linear mixed effects models) [71]. We analyzed and visualized data on pathogen load using a transformation commonly applied in this and other animal disease systems: $\log_{10}$(peak pathogen load + 1) [47,74], which allows for easier visualization and reduced skew in a variable that can range across multiple orders of magnitude. Data for this study are available at https://doi.org/10.5061/dryad.v9s4mw71r [75]. We created the map for Fig 1 using data included in the maps package in R [76].

Tolerance is standardly quantified in animals as the slope of the relationship between maximum pathogen load and maximum pathology during infection [74], frequently performed with linear regression or linear mixed effects models. Because our metric of pathology (i.e., eye score) was not a truly continuous variable (assessed at intervals of 0.5) and had both upper and lower constraints (0–3 per eye, 0–6 total), it does not fit the assumptions of such models. We therefore used the glmer function in the lme4 package with a binomial error distribution to create mixed effects logistic regression models, transforming the eye score data as follows to fit model assumptions. We first made the scale of eye score coarser, rounding up non-integer scores to the nearest integer. We then created a second, "anti-" eye score variable, calculated as 6 minus total eye score. We then modeled pathology using two numbers for each bird at each time point to define the dependent variable: the severity of potential eye pathology achieved ("eye score") and the severity of potential eye pathology not achieved ("anti-eye score"). By definition, values of these variables always summed to 6 for each individual at each sampling

point. Initial fixed effects were $\log_{10}$(peak pathogen load + 1), years of MG endemism, and their interaction. We treated years of endemism as a categorical variable with three levels because the distribution of years of MG endemism was clumped, with two populations showing little or no MG endemism (AZ, HI), two populations showing roughly 15yrs (CA, WA), and three populations showing 20-25yrs (AL, IA, VA). Because each animal contributed two data points, one pre-infection and one post-infection, and each population of origin contributed multiple individuals, the maximal model's initial random effects structure nested individual bird within population of origin. We used Akaike's Information Criterion adjusted for small sample size (AICc) [77] to compare the maximal model to simplified models (i.e., those without the interaction between $\log_{10}$(peak pathogen load + 1) and years of MG endemism; those without a random effect of population). Random effects of population were estimated near 0 (Table F in S1 Appendix), so we report results from models with only the random effect of individual in the main text. We used the Tukey method within the emmeans package to test pairwise comparisons of model estimates (i.e., differences between categories of MG endemism) [78]. For visualization purposes, we took the predictions based on the fixed effects of these models (constrained from 0–1) and multiplied by 6 to conform to the range of the original data (0–6).

Because this analysis of tolerance utilizes binomial logistic models, interpretation is slightly different than when using more familiar linear or polynomial models [74]. Specifically, we are interested in whether pathology increases with pathogen load in a different pattern in populations with longer coevolutionary histories with MG. Logistic regressions can describe such patterns without significant interaction effects between pathogen load and years of endemism. Essentially, a right-shift in the logistic curve among populations with a longer co-evolutionary history with the pathogen (a 'main' effect, not an interaction effect) will indicate that at higher pathogen loads, these populations show lower pathology and thus, higher tolerance.

Importantly, the interpretation of our tolerance results remains the same when using linear and non-linear mixed effects models that treat eye score as a continuous and/or non-constrained variable (Table G in S1 Appendix). Such models have been used in this system in the past [10,31], suggesting that despite the analysis techniques reported in the main text, our results are consistent with prior results on tolerance among house finch populations.

Asymptomatic infections were defined as those in which animals showed no eye score at any point and a pathogen load greater than 0 on at least one sampling day. We analyzed data from infections with the lower-virulence and higher-virulence isolate separately. For each isolate, we tested for overall differences in the proportion of asymptomatic infections among categories of years of MG endemism using $\chi^2$ tests. We then performed pairwise comparisons between categories of MG endemism with additional $\chi^2$ tests, adjusting p-values for multiple comparisons using the Holm-Bonferroni method [79]. Error bars were created using the Wilson method in the prevalence package [80] in R.

To test for differences in resistance (quantified as reductions in maximum pathogen load), we ran separate linear mixed effects models for the two MG isolates, each including $\log_{10}$(maximum pathogen load + 1) as the dependent variable, with fixed effects of years of MG endemism (for models of both isolates) and dose (for models of the lower-virulence isolate only), with population as a random effect in each. Pairwise comparisons between categories of MG endemism were performed as for models of tolerance.

**Dryad DOI**

https://doi.org/10.5061/dryad.v9s4mw71r [75]

## Results

### The evolution of tolerance in house finches

House finch populations with a longer history of MG endemism showed higher tolerance, manifested by less severe conjunctivitis at a given pathogen load (Fig 2). Specifically, for the lower-virulence, evolutionarily basal isolate, the number of years that MG has been endemic in a given population (categorized as 0–10 years, 10–20 years, or 20–25 years of pathogen endemism) was a strong predictor of the relationship between maximum conjunctivitis score and $\log_{10}$(maximum pathogen load + 1) (Fig 2A; Table H in S1 Appendix; generalized linear mixed-effects model: years MG endemic: Wald $\chi^2$ = 32.0, df = 2, p < 0.001). This pattern also held true for the higher-virulence, evolutionarily derived isolate (Fig 2B; Table H in S1 Appendix; years MG endemic: Wald $\chi^2$ = 26.7, df = 2, p < 0.001). In both cases, populations with 20–25 years of MG endemism were more tolerant (shallower increase in pathology with pathogen load) than populations with either 10–20 or 0–10 years of MG endemism, while we found no differences in tolerance between the latter two categories (Fig 2 and Table I in S1 Appendix). Similar patterns arise when data are analyzed using individual population as a fixed effect, instead of years of MG endemism, with higher tolerance among those populations with longer co-evolutionary histories with the pathogen (Fig B and Table J in S1 Appendix).

Populations that differed in their history of MG endemism also differed in their percentage of asymptomatic infections (i.e., infections where birds had a non-zero maximum pathogen load but a maximum eye score of 0) when inoculated with the basal, lower-virulence MG isolate (Fig 3A; $\chi^2$ = 7.24, $p$ = 0.027, N = 140 birds–one bird showed no pathogen load following inoculation, so was excluded from the analysis of asymptomatic infections). However, only populations with 10–20 years of MG endemism showed strong evidence of a higher percentage of asymptomatic infections than populations with 0–10 years of MG endemism ($\chi^2$ = 7.31, Holm-Bonferroni-adjusted $p$ = 0.021; 20–25 years vs. 10–20 years: $\chi^2$ = 1.57, Holm-Bonferroni-adjusted $p$ = 0.21; 20–25 years vs. 0–10 years: $\chi^2$ = 2.94, Holm-Bonferroni-adjusted $p$ = 0.17). When infected with the evolutionarily derived, higher-virulence isolate, the percentage of individuals with asymptomatic infections did not differ with time since MG-endemism (Fig 3B; $\chi^2$ = 0.51, $p$ = 0.78, N = 61 birds).

We found less consistent evidence for effects of MG endemism on resistance (measured as lower maximum pathogen loads for a given dose or isolate; x-axis, Fig 2). Among birds treated with the low-dose, lower-virulence isolate, differences in resistance appeared most pronounced between populations with 10–20 years of MG endemism and those with 0–10 years of endemism (Fig 2A; linear mixed effects model: years MG endemic: $F_{2,3.95}$ = 2.33, $p$ = 0.21; years MG endemic x dose: $F_{2,131}$ = 5.5, $p$ = 0.005; Tukey method for comparing estimates: 10–20 years vs 0–10 years: $p$ = 0.02; 10–20 years vs 20–25 years: $p$ = 0.22; 20–25 years vs. 0–10 years: $p$ = 0.18). This stands in contrast to tolerance, which was highest in populations with 20–25 years of endemism. Pairwise comparisons revealed no pronounced differences in resistance based on MG endemism within the high-dose, lower-virulence treatment (Fig 2A; all Tukey: p > 0.95) or the low-dose, higher-virulence isolate (Fig 2B; LM: MG endemism: $F_{2,3.1}$ = 0.31, p = 0.75).

### Mechanisms of tolerance

We identified 15,400 genes in our *de novo* reconstructed house finch transcriptome. When we compared MG-infected and non-infected birds from the same population group (i.e., more- or less-tolerant; n = 2 populations per group), 21 genes were upregulated and 18 genes were downregulated (39 genes total, Fig 4) solely among MG-infected birds in more-tolerant populations

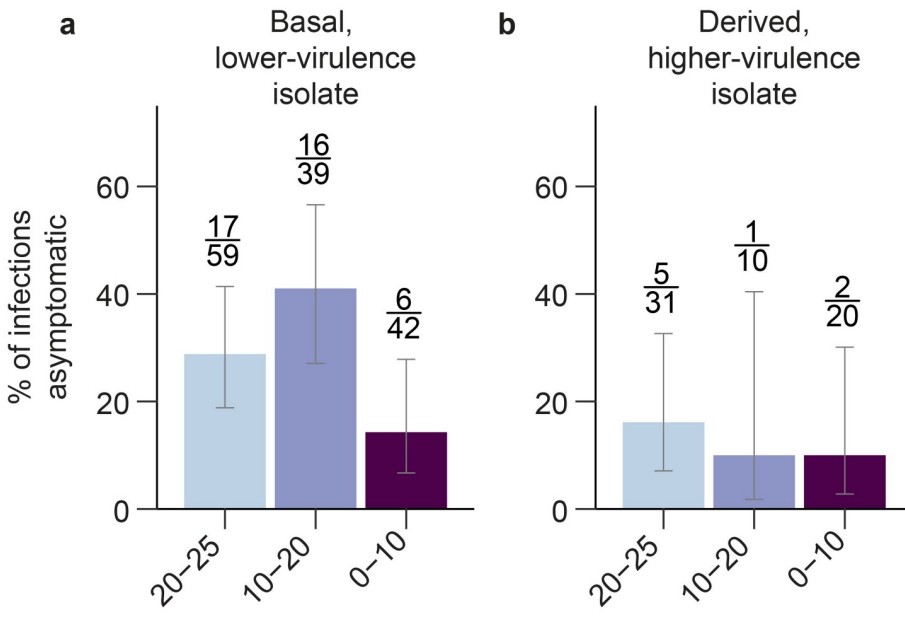

**Fig 3. The percentage of individuals with asymptomatic infections differed between populations when birds were experimentally infected with a lower-virulence MG isolate (a), a difference that was most pronounced between populations with 10–20 years and 0–10 years of MG endemism.** When birds were experimentally infected with a higher virulence isolate (b), there were no strong differences in the percentage of asymptomatic infections based on years of MG endemism. Sample sizes are included above bars, expressed as the number of asymptomatic infections/all successful infections. Only one experimentally inoculated animal showed no pathogen load following inoculation and was removed from the analysis. Error bars show 95% confidence interval estimates derived using Wilson scores in the 'prevalence' package in R.

(IA, VA). In contrast, 225 genes were upregulated and 146 were downregulated (371 genes total, Fig 4) solely among MG-infected birds in less-tolerant populations (AZ, HI). Finally, 50 genes were differentially expressed in MG-infected birds in both population groups (Fig 4). Of these, 45 were upregulated and five downregulated in MG-infected birds in less-tolerant populations, while 43 were upregulated and seven downregulated in MG-infected birds in more-tolerant populations (a full list of differentially expressed genes is provided in S2 Data).

In our enrichment analyses, a number of Gene Ontology BP (GO:BP) terms were significantly enriched (corrected p-value <0.05), predominantly for our less-tolerant populations (Table E in S1 Appendix). There were 109 significantly enriched GO:BP terms for our less-tolerant populations and eight for our more-tolerant populations. For the less-tolerant populations, the most significantly enriched terms were cell cycle terms (GO:0007049) such as mitotic cell cycle process and chromosome segregation. Many terms related to immune response were also significantly enriched for less-tolerant populations, including inflammatory response (GO:0006954), response to cytokines (GO:0034097), neutrophil migration (GO:1990266), mononuclear cell migration (GO:0071674), and humoral immune response (GO:0006959) (Fig 5). For more-tolerant populations, most significantly enriched terms were cell cycle (GO:0007049) and metabolic process terms (GO:0008152) (Table E in S1 Appendix). No immune response terms were significantly enriched for more-tolerant populations (Table E in S1 Appendix), although these populations had some differentially expressed genes in immune response terms (Fig 5).

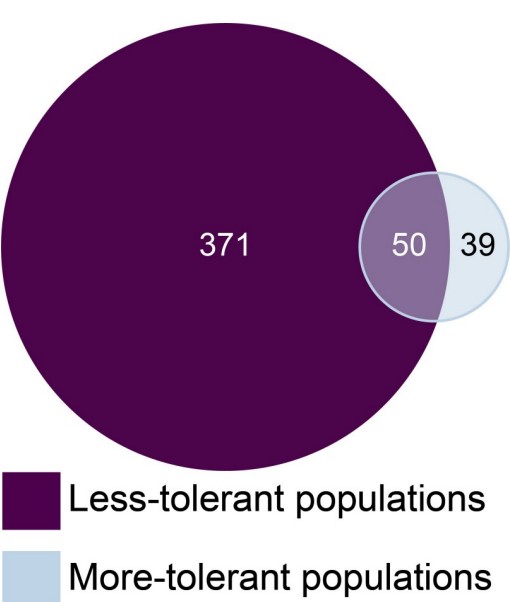

**Fig 4. In house finch Harderian glands, we identified ~90% fewer differentially expressed genes when comparing non-infected controls to MG-infected birds from populations that are more tolerant to MG (light circle) and populations that are less tolerant to MG (dark circle).** There were only 50 genes differentially expressed between non-infected controls and infected individuals in all populations.

Among genes involved in the inflammatory response, *S100A9* and *SAA1* were upregulated in response to experimental infection in birds from both less-tolerant and more-tolerant populations. Several other genes involved in inflammation were only upregulated during infection in birds from less-tolerant populations, including *S100A12*, *CCL20*, *ECM1*, *SELE*, *SELP*, *PBK*, *TNC*, *THBS1*, *CXCL8*, *CYBB*, and *CSF3R*. In terms of the humoral immune response, *TF*, *DMBT1*, *S100A9*, *S100A12*, and *IGHA1* were upregulated in birds from less-tolerant populations, while *SPINK5* and *S100A9* were upregulated and *TF* was down-regulated in birds from more-tolerant populations. Furthermore, two genes annotated as Immunoglobulin constant-1 set domains were upregulated in MG-infected birds from less-tolerant populations and down-regulated in MG-infected birds from more-tolerant populations compared to non-infected birds from these populations.

## Discussion

Overall, house finch populations with a longer history of MG endemism showed similarly high maximum pathogen loads yet significantly milder conjunctivitis during experimental infection with two distinct isolates of a recently emerged bacterial pathogen, compared with populations with shorter or no history of MG endemism. Further, populations with a longer history of MG endemism showed significantly fewer upregulated genes during MG infection than populations with a relatively shorter history of pathogen endemism. Together, these results suggest that tolerance to MG has evolved rapidly in free-living house finch populations, and that tolerance may be facilitated by a more-targeted local immune response early in infection.

Although phenotypic evolutionary changes can be difficult to conclusively demonstrate in wild vertebrate populations, our results strongly suggest an evolutionary component to the differences in tolerance detected across populations. While previous work suggested that resistance or tolerance may be evolving in house finches in response to MG [10,31,32,81], that work has focused on only two populations (AL and AZ) making it challenging to determine

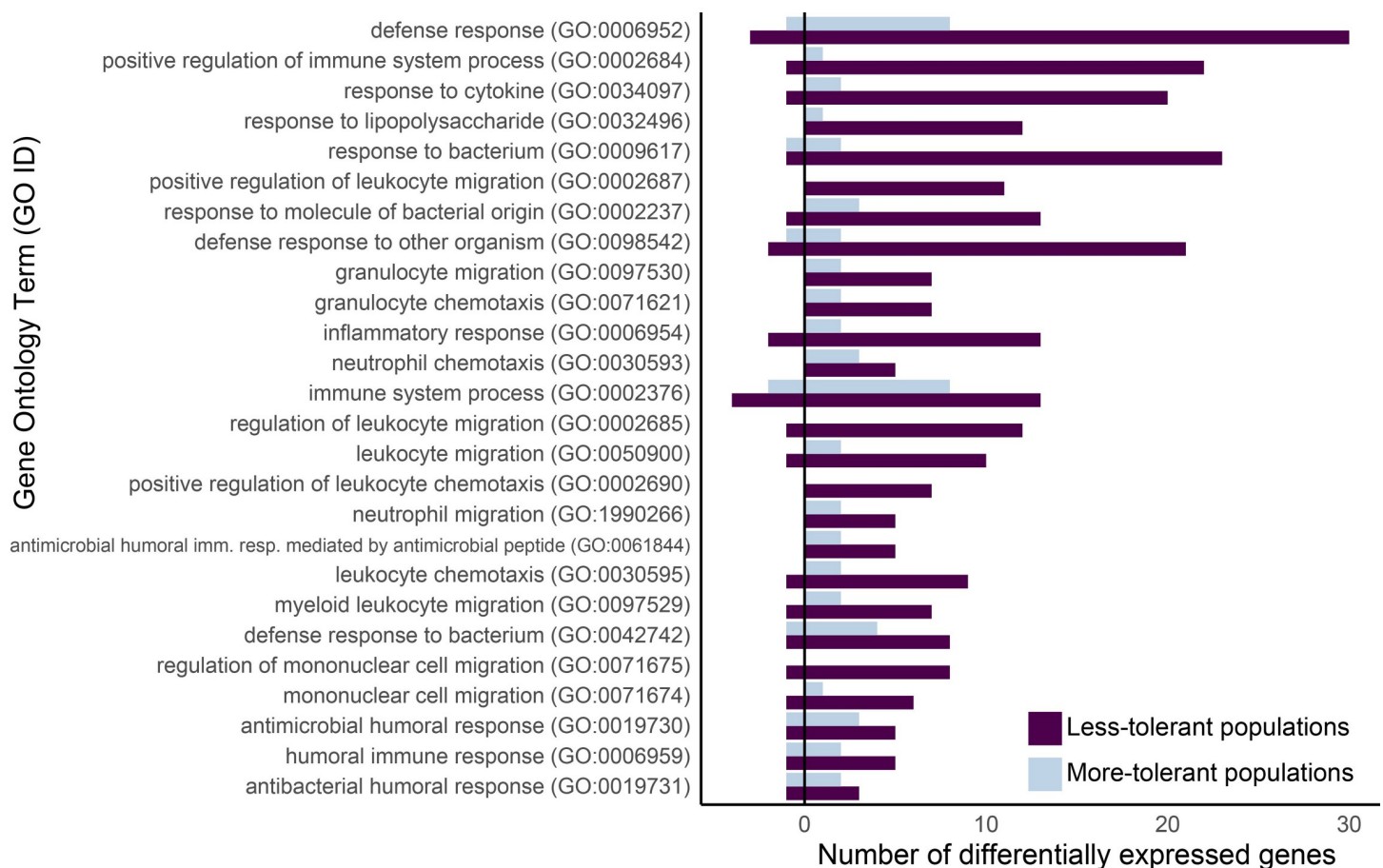

**Fig 5. Birds from less-tolerant populations (dark bars) had differentially expressed genes for a higher number of immune system Gene Ontology Biological Process terms than did birds from more-tolerant populations (light bars).** These terms are significantly enriched for birds from the less-tolerant populations but not for birds from more-tolerant populations. Genes that were upregulated in infected birds compared to non-infected birds are represented to the right of zero and genes that were downregulated in infected birds compared to non-infected birds are represented to the left of zero. Data are from 39 individuals from four populations, each with five infected and five uninfected animals (except Hawaii, as data from one uninfected animal was excluded due to low reads, see main text).

the relative contributions of MG endemism versus other confounding variables (e.g., environmental or demographic variables). Indeed, a reasonable null hypothesis might be that any two populations will differ in their response to infection, regardless of their history of MG endemism [33]. In this study, we had more than one population in both our more-tolerant and less-tolerant groups, which all differed in a number of environmental variables (e.g., climate, urbanization). By doing so, this study expands on the spatial scale of many prior studies in ecoimmunology, allowing us to better explore how tolerance varies across a pathogen's range of expansion [39]. In addition, we used MG-naïve juveniles that were captured within a few months of hatching in the wild and later exposed to MG in a common, captive environment. Thus, it is extremely unlikely that environmental differences explain the patterns of tolerance we found. Differences in demographic history or founder effects are also unlikely to explain our results. House finches are native to the western United States, while populations of house finches on the Hawaiian Islands and in the eastern US are descended from a small number of individuals introduced in ˜1870 [82] and ˜1940 [83], respectively. Thus, if demographic differences drove our results, we would expect to see a phenotypic split between all native and introduced populations. However, house finch populations on the Hawaiian Islands, which went

through a bottleneck similar to populations in the eastern US, show no tolerance to MG. In addition, when we analyze populations individually, birds from WA, which have an intermediate history of MG endemism, show a level of tolerance intermediate to those in other western and eastern populations, especially when noting the high-dose treatment (Fig Ba and Table J in S1 Appendix). Further, the level of tolerance to MG in house finch populations with 10–20 years of MG endemism (WA, CA) in this study is similar to that found in a previous study of an eastern population (the same VA population included in our current study) at a time when MG had been endemic there for ˜15 years (Fig C in S1 Appendix). Finally, there were differences in tolerance between populations regardless of whether we used a less virulent (more basal) or a more virulent (more evolved) isolate for experimental infection. Thus, in combination with previous work [10,31,32,81] our data support the hypothesis that tolerance has evolved rapidly in house finch populations—within 20–25 years, or approximately 15 host generations—since the pathogen's initial emergence and spread in the 1990s.

Despite the apparent evolution of tolerance, the increase in asymptomatic infections in populations with a longer history of MG endemism was only detected when birds were exposed to a lower virulence (more basal) isolate, not a higher virulence (more evolved) isolate. Because symptoms of infection (i.e., conjunctivitis) are important for the transmission of MG between house finches [84–86], and more-virulent MG isolates are more likely to cause infection in previously exposed individuals [40], strong selective pressures on MG (favoring increased virulence) may prevent the evolution of completely asymptomatic infections in house finches.

In the house finch-MG system, previous work has mostly explored gene expression in splenic tissue and later in infection (e.g., 14 days post-inoculation) [81,87,88], which may explain differences between gene expression patterns reported here and those from earlier work. For example, previous work using splenic tissue showed an upregulation of immune genes in birds from populations with a longer history of MG endemism, and downregulation or no change in immune genes in birds from populations with no history of MG endemism [81,87,88]. Here, we find the opposite pattern in Harderian glands: little change in immune gene expression in birds from populations with a longer history of MG endemism and an upregulation of immune genes in birds from populations with little or no history of MG endemism.

We chose to measure gene expression in the Harderian gland (an eye-associated lymphoid tissue [41]) early in infection to examine localized, innate (e.g., inflammatory) responses that are likely extremely relevant to MG pathology (conjunctivitis) and, therefore, tolerance. As we found an upregulation of many immune genes only in MG-infected birds from less-tolerant populations, selection from MG has potentially favored muting finch immune responses (e.g., inflammatory pathways) that could result in immunopathology. While two genes that positively regulate the inflammatory response (i.e., S100A9 and SAA1) [89,90] were upregulated in birds from both less- and more-tolerant populations, a number of additional pro-inflammatory genes were upregulated only in birds from less-tolerant populations. For example, a cytokine receptor (CSF3R) and three chemokines (CXCL8, CXCL14, CCL20) responsible for leukocyte trafficking were upregulated during infection only in birds from less-tolerant populations. CCL20 attracts lymphocytes and dendritic cells to mucosal lymphoid tissues [91], CXCL8 (IL-8) attracts neutrophils [92], and CXCL14 attracts a variety of leukocytes, including neutrophils and natural killer cells, and shows bactericidal activity [93]. In contrast, SPINK5, which may have an anti-inflammatory role in mucosal tissues [94,95], was upregulated only in birds from more-tolerant populations. Previous studies comparing two house finch populations also suggested that birds from populations with a history of MG-endemism have decreased expression of pro-inflammatory cytokines early in infection, compared to birds

from MG-naïve populations [31,96]. Such differences in gene expression only three days following inoculation suggest that early immune responses may play an important role in shaping tolerance throughout infection.

Although gene expression studies are an important tool for understanding the functional nature of immune response in non-model organisms, there are limitations inherent in using gene expression data. For example, as discussed here, gene expression in response to infection likely depends on the tissue used and stage of infection. Further, gene expression levels may not correlate directly with protein levels [97]. Thus, future studies that aim to fully characterize immune responses may benefit from examining both the proteome [98] and transcriptome in multiple tissues across the course of infection.

## Conclusions

Taken together, our data suggest that tolerance of infection evolved rapidly in house finches after the emergence of a novel pathogen, *M. gallisepticum*, and that a smaller number of differentially expressed genes characterize responses to infection in more-tolerant than in less-tolerant populations. Moreover, these data show that tolerance can be an important means by which hosts evolve to combat emerging pathogens. However, the evolution of completely asymptomatic infections may be prevented through the co-evolution of increased pathogen virulence if tolerance reduces pathology that is important for pathogen transmission. Thus, the evolution of tolerance can have wide-ranging consequences for both pathogen transmission and host-pathogen co-evolution in the wild.

## Supporting information

**S1 Appendix. Supplementary figures, tables, methods, and results that are referenced in the main manuscript.**
(PDF)

**S1 Data. This supplement contains a complete list of all terms that were significantly enriched using our transcriptomic data.** This includes which populations the terms are enriched for (more-tolerant or less-tolerant), whether the genes in the term were upregulated or downregulated compared to control birds, which database the term is associated with, the corrected p-value for each term, the number of genes in each term, and a list of the genes in each term.
(TXT)

**S2 Data. This supplement contains a full list of differentially expressed genes, including which population the gene is differentially expressed in (more-tolerant or less-tolerant), the annotation for the gene (if available), the log fold change of the gene compared to non-infected controls, whether the gene is upregulated or downregulated compared to controls, and the corrected p-value for each gene.**
(TXT)

## Acknowledgments

We would like to thank the many research assistants that helped us in the field and lab. We would also like the thank L. Deanovic and T. Hahn (UC-Davis), P. Hutton and K. McGraw (Arizona State University), G. Hill and W. Hood (Auburn University), and S. Goldstein, P. Howard, and J. Omick (Hawaii USDA) for their help with logistics in the field. In addition, we

are very grateful to both M. Sayadi and A. Severin (ISU Genome Informatics Facility) for their help with analyzing gene expression data.

## Author Contributions

**Conceptualization:** Amberleigh E. Henschen, Michal Vinkler, Marissa M. Langager, Rami A. Dalloul, Dana M. Hawley, James S. Adelman.

**Data curation:** Amberleigh E. Henschen, Dana M. Hawley, James S. Adelman.

**Formal analysis:** Amberleigh E. Henschen, Dana M. Hawley, James S. Adelman.

**Funding acquisition:** Rami A. Dalloul, Dana M. Hawley, James S. Adelman.

**Investigation:** Amberleigh E. Henschen, Michal Vinkler, Marissa M. Langager, Allison A. Rowley, Rami A. Dalloul, Dana M. Hawley, James S. Adelman.

**Methodology:** Amberleigh E. Henschen, Michal Vinkler, Marissa M. Langager, Allison A. Rowley, Rami A. Dalloul, Dana M. Hawley, James S. Adelman.

**Project administration:** Amberleigh E. Henschen, Michal Vinkler, Rami A. Dalloul, Dana M. Hawley, James S. Adelman.

**Resources:** Michal Vinkler, Rami A. Dalloul, Dana M. Hawley, James S. Adelman.

**Supervision:** Amberleigh E. Henschen, Michal Vinkler, Rami A. Dalloul, Dana M. Hawley, James S. Adelman.

**Validation:** Amberleigh E. Henschen, Marissa M. Langager, Allison A. Rowley, Rami A. Dalloul, Dana M. Hawley, James S. Adelman.

**Visualization:** Amberleigh E. Henschen.

**Writing – original draft:** Amberleigh E. Henschen, James S. Adelman.

**Writing – review & editing:** Amberleigh E. Henschen, Michal Vinkler, Marissa M. Langager, Allison A. Rowley, Rami A. Dalloul, Dana M. Hawley, James S. Adelman.

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
