## [Decision Letter · Decision Letter 0]

3 Feb 2023

Dear Henschen,

Thank you very much for submitting your manuscript "Rapid adaptation to a novel pathogen through disease tolerance in a wild songbird" for consideration at PLOS Pathogens. As with all papers reviewed by the journal, your manuscript was reviewed by members of the editorial board and by several independent reviewers. The reviewers appreciated the attention to an important topic. Based on the reviews, we are likely to accept this manuscript for publication, providing that you modify the manuscript according to the review recommendations.

It is important to note that one of the reviewers asked for a substantial re-do on the statistical analysis stating concerns that appear to be well founded. Please take care to address all reviewer concerns in your revision.

Sincerely,

Elizabeth A. McGraw, PhD

Academic Editor

PLOS Pathogens

Ronald Swanstrom

Section Editor

PLOS Pathogens

Kasturi Haldar

Editor-in-Chief

PLOS Pathogens

orcid.org/0000-0001-5065-158X

Michael Malim

Editor-in-Chief

PLOS Pathogens

orcid.org/0000-0002-7699-2064

Reviewer Comments (if any, and for reference):

Reviewer's Responses to Questions

**Part I - Summary**

Reviewer #1: Owing to the spatial extent of the sampling, paired with experimental pathogen challenge, this study represents a rare and important contribution to the areas of ecoimmunology and host evolution in response to novel pathogens. My comments are relatively minor, although I did have some concerns about the statistical analysis, particularly how conjunctivitis score is modeled, the potential importance of reporting random effect variance, and the robustness of the differential expression analyses.

Reviewer #2: This report presents a comprehensive analysis of both the evolution and mechanistic basis of tolerance in the well studied house finch-Mycoplasma system. The main conclusions are that populations with a longer history with Mycoplasma have higher tolerance (and resistance) than populations with a shorter history, and that this is achieved through suppressed immune responses. The study is based on an impressive data set involving common garden experiments with birds from 7 populations. The presentation is clear and the manuscript is well written. The study is a rare example of adaptation in real-time to infectious disease in a wild study system.

**Part II – Major Issues: Key Experiments Required for Acceptance**

Reviewer #1: L325: If conjunctivitis score is the dependent variable, and this ranges from 0-6 (with a maximum of six), have the authors considered using a GLMM instead? A gaussian response (LMM) could theoretically allow your predicted values to extend beyond the maximum eye score possible. A GLMM with binomial response (e.g., cbind(positive eye score, negative eye score)) would bound the model predictions within 0-6. Alternatively, have the authors considered a beta regression model to just model the response as a proportion (e.g., 4/6 eye score)? A poisson response would also be appropriate, although technically that could produce eye scores above 6.

L316: The analysis of molecular pathways is slightly vague. Did the authors perform a formal enrichment analysis (e.g., g:Profiler or another similar method)? This would be an ideal method to robustly identify differently up/downregulated pathways.

L356: Can the authors report variance estimates for the random effects (i.e., individual ID and population)? This could provide important insights into repeatability and population effects.

Figure 2: The fact that the quadratic slope drops below eye score of 0 indicates that the LMM is producing impossible fitted values and why a GLMM with a more appropriate response should be used (e.g., modeling eye score as counts of successes to failures, a proportion, true counts, etc; see above). The same issue is evident in Figure S3.

L472-479 and Figure 4/5: As it currently reads, the pathway analysis doesn’t demarcate significantly up/downregulated pathways, which should be obtainable with a formal enrichment analysis.

Reviewer #2: My only major comment is that it is not clear if experiments with all populations were performed at the same time. I guess not, since different populations were captured in different years (L 242-). Would be good with a (supplementary) table specifying which populations were studied in which year, and at which institute. I also wonder if there were any differences in effects of infection (load or symptoms) between institutes and years. I understand that the logistics were necessarily complicated and there is no need for new experiments, but you need to show that the results are not an artefact of non-ideal experimental design (e.g. that experiments with different populations were performed at different institutes or different years, or that time in captivity prior to experiments differed between populations).

**Part III – Minor Issues: Editorial and Data Presentation Modifications**

Reviewer #1: L88: Because MG is defined here, you could move the abbreviation on L90 to here instead.

L133-134: The authors largely focus on using a space-for-time approach, but I think the authors could still better emphasize the importance of the spatial aspects of the study. Multi-site comparisons of immunity in wild hosts are rare and often have low spatial replication (e.g., see a discussion by Becker et al. 2020 JAE). Seven populations sampled across the geographic range of this species is more than most ecoimmunology work and allows the authors to capitalize on a wide range of immunogenetic variation (given the common garden approach).

L142: I think it would make more sense to simply replace “RNAsequencing” with “transcriptomics”

L197: Here and elsewhere, replace “RNASeq” with “RNA-Seq”

L307: “EdgeR” should be “edgeR” to match the R package name.

L385-388 and L389-393: Rather than report the F statistics and p values for each term in the LMM, the authors could consider moving this into an ANOVA table in the main text or supplement and focus on just reporting the significant interactions here (as the main effects are meaningless to interpret with the significant interaction term).

L396: See the earlier comment about reporting variance estimates for individual and population.

Figure 2: It would be nice to see confidence intervals around each of the fitted quadratic slopes here; the authors should consider adding these via bootstrapping or another method for LMMs.

Figure 3: It would be nice to see confidence intervals around each proportion (e.g., consider using the prevalence package to derive Wilson intervals for smaller sample sizes)

L541-545: These are good points; is there any evidence for spatial variation based on these described factors in immune phenotypes (e.g., more classic ecoimmunology markers) in the HOFI system? Additionally, reporting random effect variance for individual and population may be another way to slightly “get at” population effects relative to the fixed effect of interest (years endemic).

L581: Replace “RNAseq” with “RNA-Seq”

L618: One other point the authors may with to briefly discuss is how the expression of immune genes observed here may (or may not!) translate into functional response, given known differences between gene expression and protein abundance. Proteomics may be an applicable approach to mention for future work in this regard (e.g., see Heck & Neely 2020 JPR).

Reviewer #2: Even if there are no differences or effects of timing of experiments or where experiments were performed (see major comment above), the authors should acknowledge that the study is not a proper common garden experiment (which should have involved breeding for at least one generation in captivity to eliminate maternal effects) so that differences between populations could be a result of environmental effects rather than adaptation.

L 72: Ref 5 (Bonneaud) is not about bats. Also, I thought defence against WNS in bats (I assume this is what you have in mind) was a case of tolerance, not resistance.

L 100: “of expected”? “Reduce by 40%” better?

L 268: Briefly describe the function of the Harderian gland, and what organ it corresponds to in mammals (if any).

L285: Very deep sequencing, especially considering birds have smaller genomes than for ex mammals!

L290: Is there no reference genome for this species?

L 420: Asymptomatic means score=0 I assume, but perhaps best to specify?

L 472-: The pathway analysis is a bit unconventional in that you analyze number of DEG in different pathways, rather than which pathways were enriched for DEG; why was the analysis performed in this way?

L 579: You cannot really say anything about which pathways are suppressed, so delete “less-specific…early”.

L 602-609: Not really meaningful to discuss previous studies in so much detail, so condense this.

PLOS authors have the option to publish the peer review history of their article (what does this mean?). If published, this will include your full peer review and any attached files.

Reviewer #1: No

Reviewer #2: No

Figure Files:

Data Requirements:

Reproducibility:

References:

---

## [Editor Report · Decision Letter 1]

8 May 2023

Dear Henschen,

We are pleased to inform you that your manuscript 'Rapid adaptation to a novel pathogen through disease tolerance in a wild songbird' has been provisionally accepted for publication in PLOS Pathogens.

Best regards,

Elizabeth A. McGraw, PhD

Academic Editor

PLOS Pathogens

Ronald Swanstrom

Section Editor

PLOS Pathogens

Kasturi Haldar

Editor-in-Chief

PLOS Pathogens

orcid.org/0000-0001-5065-158X

Michael Malim

Editor-in-Chief

PLOS Pathogens

orcid.org/0000-0002-7699-2064
---

## [Editor Report · Acceptance letter]

7 Jun 2023

Dear Henschen,

We are delighted to inform you that your manuscript, "Rapid adaptation to a novel pathogen through disease tolerance in a wild songbird," has been formally accepted for publication in PLOS Pathogens.

Best regards,

Kasturi Haldar

Editor-in-Chief

PLOS Pathogens

orcid.org/0000-0001-5065-158X

Michael Malim

Editor-in-Chief

PLOS Pathogens

orcid.org/0000-0002-7699-2064